# Deep Projective Rotation Estimation through Relative Supervision

**Brian Okorn,**[*] **Chuer Pan,**[*] **Martial Hebert, David Held**
Robotics Institute, School of Computer Science
Carnegie Mellon University, United States
{bokorn, chuerp, mhebert, dheld}@andrew.cmu.edu

**Abstract:** Orientation estimation is the core to a variety of vision and robotics tasks such as camera and object pose estimation. Deep learning has offered a way to develop image-based orientation estimators; however, such estimators often require training on a large labeled dataset, which can be time-intensive to collect. In this work, we explore whether self-supervised learning from unlabeled data can be used to alleviate this issue. Specifically, we assume access to estimates of the relative orientation between neighboring poses, such that can be obtained via a local alignment method. While self-supervised learning has been used successfully for translational object keypoints, in this work, we show that naively applying relative supervision to the rotational group $SO(3)$ will often fail to converge due to the non-convexity of the rotational space. To tackle this challenge, we propose a new algorithm for self-supervised orientation estimation which utilizes Modified Rodrigues Parameters to stereographically project the closed manifold of $SO(3)$ to the open manifold of $\mathbb{R}^3$, allowing the optimization to be done in an open Euclidean space. We empirically validate the benefits of the proposed algorithm for rotational averaging problem in two settings: (1) direct optimization on rotation parameters, and (2) optimization of parameters of a convolutional neural network that predicts object orientations from images. In both settings, we demonstrate that our proposed algorithm is able to converge to a consistent relative orientation frame much faster than algorithms that purely operate in the $SO(3)$ space. Additional information can be found on our website.

## 1 Introduction

Pose estimation is a critical component for a wide variety of computer vision and robotic tasks. It is a common primitive for grasping, manipulation, and planning tasks. For motion planning and control, estimating an object's pose can help a robot avoid collisions or plan how to use the object for a given task. The current top performing methods for pose estimation use machine learning to estimate the object's pose from an image; however, training these estimators tends to rely on direct supervision of the object orientation [1, 2, 3]. Obtaining such supervision can be difficult and requires either time-consuming annotations or synthetic data, which might differ from the real world. In this work, we explore whether self-supervised learning can be used to alleviate this issue by training an object orientation estimator from unlabeled data. Specifically, we assume that we can estimate the relative rotation of an object between neighboring object poses in a self-supervised manner. Such relative supervision can be easily obtained in practice, for example through a local registration method such as Iterative Closest Point (ICP) [4] or camera pose estimation.

Relative self-supervision has been previously used for representation learning in estimating translational keypoints [5, 6, 7]. These methods use only relative supervision to ensure that the keypoints are consistent across views of the object, and do not directly supervise the keypoint locations. In this work, we explore whether such relative self-supervision can similarly be used in estimating object orientations. We show that naively applying such relative supervision to rotations on the $SO(3)$ manifold will often fail to converge. Unlike self-supervised learning of translational keypoints, the

---

[*]Equal Contribution

6th Conference on Robot Learning (CoRL 2022), Auckland, New Zealand.

rotational averaging problem [8] is inherently non-convex, with many local optima. While there exist global optimization algorithms which jointly optimize all pairs of rotations for this problem [9, 10], they are not easily integrated into the iterative, stochastic gradient descent methods used to train neural network-based pose estimators.

To address this issue, we propose a new algorithm, Iterative Modified Rodrigues Projective Averaging, which uses Modified Rodrigues Parameters to map from the closed manifold of $SO(3)$ to the open space of $\mathbb{R}^3$. In doing so, we obtain faster convergence with a lower likelihood of falling into local optima. Our experiments show that our method converges faster and more consistently than the standard $SO(3)$ optimization and can easily be integrated into a neural network training pipeline. Additionally, in the Appendix A, we include an intuitive theoretical example describing how, while not all local optima are removed, the dimensionality of a set of problematic configurations is greatly reduced when optimizing using our algorithm, as compared to optimizing in the space of $SO(3)$.

The primary contributions of this work are:

- We propose a new algorithm, Iterative Modified Rodrigues Projective Averaging, which is an iterative method for learning rotation estimation using only relative supervision and can be applied to neural network optimization.

- We empirically investigate the convergence behavior of our algorithm as compared to optimizing on the $SO(3)$ manifold.

- We demonstrate that our algorithm can be used to train a neural network-based pose estimator using only relative supervision.

## 2 Related Work

**Averaging and Consensus Estimation:** Consensus methods, sometimes referred to as averaging methods, have a long history of research. The goal of these methods is, given a distributed set of estimates, to produce a consistent prediction of a value using relative information. While there are iterative algorithms with good convergence properties in Euclidean space [11, 12, 13, 14, 15], optimizing over the closed manifold of $SO(3)$ can be more difficult, as the region is non-convex, with many local minima. Hartley et al. [8, 16] describe several methods of finding a consistent set of rotations, though their convergence is similarly not guaranteed outside of a radius $r \leq \frac{\pi}{2}$ ball in $SO(3)$. Wang and Singer [10] find an exact solution to this problem, using a combination of a semidefinite programming relaxation and a robust penalty function. More recently, Shonan Rotation Averaging [9] shows that projecting to higher dimensional spaces allows for the recovery of a globally optimal solution using semidefinite programming. Chatterjee and Govindu [17, 18] use iterative re-weighted least-squares to recover a global optimal solution using global error estimates. Shi and Lerman [19] extends this work, using cycle consistency and message passing. These solutions require global error estimates or semidefinate programming, which are incompatible with the stochastic gradient descent methods used to train neural networks.

**Supervised Orientation Estimation:** Past work has explored using a neural network to predict an object's orientation. Traditionally, these methods rely on supervising the rotations using a known absolute orientation, whether in the form of quaternions [20, 1, 21], axis-angle [22], or Euler angles [23]. More recently, 6D [24, 2], 9D [25], and 10D [26] representations have been developed for continuity and smoothness. Recently, Terzakis et al. [27] introduced Modified Rodrigues Parameters, a projection of the unit quaternion sphere $\mathbb{S}^3$ to $\mathbb{R}^3$ used in attitude control [28], to a range of common computer vision problems. Terzakis et al. [27] does not, however, address the unique problems found in the rotation averaging problem.

Recently, there has been research into mapping the Riemannian optimization to the Euclidean optimization used for network training [29, 30, 31, 32, 33]. These methods focus on applying tangent space gradients from losses in 3D transformation groups. Specifically, Projective Manifold Gradient Layer [29] ensures that the gradients take into account any projection operations, such that the gradients point towards the nearest valid representation in the projection's preimage. While this does map the Riemannian optimization into a Euclidean problem, it does not solve the problems caused by the closed manifold of $SO(3)$, as this does not alter the underlying topology of this manifold.

## 3 Problem Definition

We formally describe the problem of self-supervised orientation estimation below. We assume that we are given a set of inputs observations $\{I_1, \ldots, I_N\}$, of an object where, in each input observation $I_i$, the object is viewed from an unknown orientation $R_i$. These inputs could be in the form of images, point clouds, or some other object representation. While we do not know the absolute object orientations $R_i$ in any reference frame, we assume that we do know a subset of the relative rotations $R_j^i$, possibly from a local registration method like ICP, between the object in images $I_j$ and $I_i$, such that $R_i = R_i^j R_j$. Our goal is to learn a function $f(I_i)$ that estimates an orientation of the object in each image, $f(I_i) = \hat{R}_i$ that minimizes the pairwise error between all input pairs and their ground truth relative rotations, with respect to the geodesic distance metric $d(R_i, R_j) = \|\log(R_i^\top R_j)\|^2$. Given a set of rotations $\mathcal{R} = \{R_1, \ldots, R_N\}$, the core optimization objective is thus:

$$\min_{\hat{R}_i, \hat{R}_j \in \mathcal{R}} \sum_{i,j} d(\hat{R}_i, R_i^j \hat{R}_j) \tag{1}$$

Note that this optimization does not have a unique solution, since the solution $\hat{R}_i := SR_i, \forall i$ minimizes this error for any constant rotation $S$. In many robotics tasks, relative rotations can be accurately estimated only when their magnitude is small as many registration algorithms, such as ICP, requires a good initialization near the optimum. Following this observation, we assume that we can only accurately supervise relative rotations when they are small in magnitude. This leads to a local neighborhood structure where each rotation $R_i$ is connected to $R_j$ only in a local neighborhood around $R_i$, when $d(R_i, R_j) < \epsilon$, and the set of all $R_j$'s connected to $R_i$ form the neighborhood set of $\mathcal{N}_i$. While the algorithms described in this manuscript do not rely on this angle $\epsilon$, it can be scaled as needed based on the accuracy of the relative rotation estimation method (e.g. ICP, etc).

Our eventual goal is to represent the function $f(I_i) = \hat{R}_i$ as a neural network. Thus, we restrict the methods with which we compare to iterative methods that are updated using only a sampled subset of the rotations (as opposed to methods that perform a global optimization over the entire set of rotations $\{R_1, \ldots, R_N\}$). This requirement is to match the conditions required by stochastic gradient descent, the primary method of training neural networks.

## 4 Baselines

**Preliminaries.** The 3D rotational space of $SO(3) \triangleq \{R \in \mathbb{R}^{3 \times 3} : R^\top R = \mathbb{I}_{3 \times 3}, \det(R) = 1\}$ is a compact matrix Lie group, which topologically is a compact manifold. Due to the compactness of the $SO(3)$ manifold, there exist configurations of pairs of points where multiple, non-unique geodesically minimal paths exist between them; for instance, there are two unique geodesically minimal paths for a pair of antipodal points on a circle, and there are infinitely many for a pair of antipodal points on a sphere. This is not the case in an open manifold like the 3D Euclidean space of $\mathbb{R}^3$, over which there exists a unique geodesically minimal path between any arbitrary pair of points. The distinction in compactness between the 3D rotational space of $SO(3)$ and 3D Euclidean space makes optimization over $SO(3)$ more ill-conditioned than over the space of $\mathbb{R}^3$. This results in the optimization over the rotational space being non-convex. These properties of the $SO(3)$ manifold will affect the convergence of self-supervised orientation estimation, which we discuss below.

While self-supervised learning for objects translation, specifically in the form of object keypoints [5, 6, 7], has shown great success, in this work, we show that naively applying such an iterative self-supervised formulation to the rotational group $SO(3)$ will often fail to converge. Below we discuss two approaches to self-supervised orientation estimation in $SO(3)$.

**Quaternion Averaging:** A standard objective in rotation estimation is to minimize the geodesic distance between a predicted unit quaternion and its corresponding ground-truth orientation [34, 8], $\theta = \arccos(2\langle \hat{q}_i, q_{gt} \rangle^2)$ where $\hat{q}_i$ is the predicted orientation for image $i$ and $q_{gt}$ is the ground-truth orientation. An objective function is often defined to directly minimize this geodesic distance [34].

In our task, defined above (Section 3), we are given the relative rotation $q_i^j$ between some pairs of rotations $q_i$ and $q_j$. Using this relative supervision, we can use the geodesic distance between a sampled estimate, $\hat{q}_i$, its desired relative position with respect to a sampled neighbor and a known

relative rotation $q_i^j$, $\tilde{q}_i = q_i^j \otimes \hat{q}_j$, leading to the loss

$$\mathcal{L}_q = 1 - \langle \hat{q}_i, q_i^j \otimes \hat{q}_j \rangle^2 \tag{2}$$

where $\otimes$ denotes the quaternion multiplication. Note that this loss is monotonically related to the geodesic distance when using unit quaternions, while avoiding the need to compute an $\arccos$.

$SO(3)$ **Averaging:** To optimize the rotations with respect to the non-Euclidean geometry of the rotational manifold of $SO(3)$, one approach is described by Manton [35]. Each orientation is iteratively updated in the tangent space using the logmap of $SO(3)$ and projected back to $SO(3)$ using the exponential map. Specifically, we can take the gradient of the loss

$$\mathcal{L}_{SO(3)} = \left\| \log \left( R_i^\top R_i^j R_j \right) \right\|^2 \tag{3a} \qquad \nabla_{\hat{r}_i} \mathcal{L}_{SO(3)} = r_\Delta = \log \left( R_i^\top R_i^j R_j \right) \tag{3b}$$

which gives the update step $\hat{R}_i \leftarrow \hat{R}_i \exp(\gamma r_\Delta)$, where $\gamma$ is the learning rate and $\log$ is the logmap of $SO(3)$. When optimizing the full set of orientations, this algorithm can fall into local optima due to the closed nature of the space which allows any orientation to be reached by more than one unique straight paths, as the space wraps around on itself.

# 5 Method

We propose an alternative that projects the optimization to an open image and optimizes the distances in that space. Specifically, we use the Modified Rodriguez Projection to minimize the relative error between neighboring poses in $\mathbb{R}^3$. We provide experiments in Section 6 that show that self-supervised orientation estimation using Modified Rodriguez Projection converges much faster than self-supervised orientation estimation in $SO(3)$, with theoretic analysis of an illustrative example available in the Appendix A.

**Iterative Modified Rodrigues Projective Averaging**

As mentioned previously, optimizing on a closed space, such as $SO(3)$ or $\mathbb{S}^3$ can be problematic, since the relative distance between two points can eventually be minimized by moving them in the exact opposite direction of the minimum path between them. To alleviate this issue, we would like to instead perform self-supervised learning in an open space, where this symmetry is broken. This can be done using Modified Rodrigues Parameters (MRP) [36, 27]. MRP is the stereographic projection of the closed manifold of the quaternion sphere $\mathbb{S}^3$ to $\mathbb{R}^3$, and has been widely used in attitude estimation and control [28]. In combining this projection with the mapping between $SO(3)$ and $\mathbb{S}^3$, this projection can be used to optimize ro-

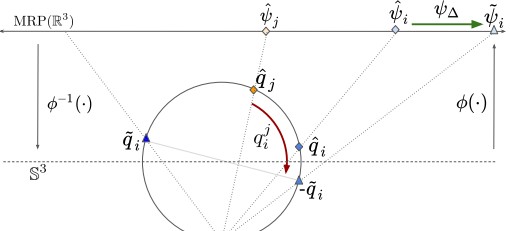

Figure 1: Projection of relative supervision, $q_i^j$, shown in red, between rotations $\hat{q}_j := \phi^{-1}(\hat{\psi}_j)$ and $-\tilde{q}_i$, into the MRP space update, $\psi_\Delta$, shown in green. While $\tilde{q}_i$ could have been selected as the the goal rotation, it would have induced a much larger movement in the projected space.

tations. We define a unit quaternion $q = [\rho \quad \nu] \in \mathbb{S}^3 \triangleq \{x \in \mathbb{R}^4 : \|x\| = 1\}$, where $\rho \in \mathbb{R}$ defines the scalar component and $\nu \in \mathbb{R}^3$ defines the imaginary vector component of the unit quaternion. The projection operator $\phi(q) = \psi \in \mathbb{R}^3$ and its inverse $\phi^{-1}(\psi) = q \in \mathbb{S}^3$ are given by [36, 27] where $\psi = \phi([\rho \quad \nu]) = \frac{\nu}{1+\rho}$ and $[\rho \quad \nu] = \phi^{-1}(\psi) = \left[ \frac{1-\|\psi\|^2}{1+\|\psi\|^2} \quad \frac{2\psi}{1+\|\psi\|^2} \right]$. Given this projective orientation space, we need to map our relative rotation $R_i^j$ into the projective space in order to use these relative rotations for the self-supervised learning task.

This projection is required, as the relative supervision is in $SO(3)$, and the direction and magnitude of this relative measurement are distorted differently in different regions of the projective MRP space. Given a pair of estimated projected rotations $\hat{\psi}_i := \phi(\hat{q}_i)$ and $\hat{\psi}_j := \phi(\hat{q}_j)$, we project $\hat{\psi}_j$ back to a unit quaternion $\phi^{-1}(\hat{\psi}_j) = \hat{q}_j \in \mathbb{S}^3$ and rotate it according to $R_i^j$, $\tilde{q}_i = q_i^j \otimes \hat{q}_j$, where $\otimes$ is quaternion multiplication and $q_i^j$ is the quaternion form of $R_i^j$. The resulting unit quaternion $\tilde{q}_i$ is

then projected back into the Modified Rodrigues Parameter space, $\tilde{\psi}_i$. A simplified visual analogy of this process is shown in Figure 1.

While this relative rotation could be applied and projected at either the sampled point $\hat{\psi}_i$, or the neighboring location $\hat{\psi}_j$, we select the neighboring location $\hat{\psi}_j$, as it does not require us to compute gradients through the forward or inverse projections $\phi(\cdot)$ and $\phi^{-1}(\cdot)$, respectively. This projected rotation $\tilde{\psi}_i$ represents the value $\hat{\psi}_i$ should hold, relative to the current predicted rotation $\hat{\psi}_j$. It should be noted that $\phi(q) \neq \phi(-q)$, while $q$ and $-q$ represent the same rotation. In terms of the projective space, this means that the sign of $\tilde{q}_i$ matters. To remove this ambiguity, we select the nearest projection to $\hat{\psi}_i$ in the projective MRP space. It should be noted that this is different from selecting the closer antipode on $\mathbb{S}^3$, as the large deformations found near the south pole[2] can cause the nearer antipode in $\mathbf{S}^3$ to be further in MRP space. In contrast, if we were to select a consistent sign for the scalar component $\tilde{q}_i$, for example ensuring the scalar component is always positive, a small change in $\hat{\psi}_j$ can cause large changes in $\tilde{\psi}_i$ when $\phi^{-1}(\hat{\psi}_j)$ is near the equator of $\mathbb{S}^3$. While this change is required to stabilize our optimization, it does add some ambiguity to the direction of optimization. However, the directions to each of the projected locations, $\phi(\tilde{q}_i)$ and $\phi(-\tilde{q}_i)$, can only be anti-parallel (pulling in exactly opposite directions) if $\tilde{\psi}_i - \hat{\psi}_i$ intersects the origin in $\mathbb{R}^3$.

The loss with respect to a given estimate, $\hat{\psi}_i$, can then be written as the $l_2$ distance between its current value and the projected relative location, $\tilde{\psi}_i$, relative to a given neighbor, $\hat{\psi}_j$:

$$\mathcal{L}_{\Psi+} = \left\| \hat{\psi}_i - \phi(\tilde{q}_i) \right\|^2 \quad \text{(4a)} \quad \mathcal{L}_{\Psi-} = \left\| \hat{\psi}_i - \phi(-\tilde{q}_i) \right\|^2 \quad \text{(4b)} \quad \mathcal{L}_{\Psi} = \min(\mathcal{L}_{\Psi-}, \mathcal{L}_{\Psi+}) \quad \text{(4c)}$$

where we recall that, $\tilde{q}_i = q_i^j \otimes \hat{q}_j$, and $\hat{q}_j = \phi^{-1}(\hat{\psi}_j)$.

Note that, while $\hat{\psi}_j$ is a predicted value, we do not pass gradients through it, allowing it to anchor the update to a consistent orientation. The gradient update[3] is then given by:

$$\nabla_{\hat{\psi}_i} \mathcal{L}_{\Psi} = \psi_{\Delta} = \begin{cases} \hat{\psi}_i - \phi\left(\tilde{q}_i\right), & \text{if } \mathcal{L}_{\Psi+} < \mathcal{L}_{\Psi-} \\ \hat{\psi}_i - \phi\left(-\tilde{q}_i\right), & \text{otherwise} \end{cases} \quad (5)$$

Additionally, a maximum gradient step, $\eta$, in the projective space is imposed, $\psi_{\Delta} \leftarrow \eta \frac{\psi_{\Delta}}{\|\psi_{\Delta}\|}$, if the gradient exceeds a defined amount. This prevents extremely large steps from being taken, as the projective transform can distort the space.

# 6 Experiments

Next, we perform experiments to show that our method converges faster and more consistently than the alternative approaches. Our empirical results are grouped into two settings: (1) direct optimization of randomly generated rotations, Section 6.1, and (2) optimization of the parameters of a convolutional neural network using synthetically rendered images, Section 6.2. In both cases, relative orientations between elements in a neighborhood are provided. We show Iterative Modified Rodrigues Projective Averaging is able to converge faster and more often than alternative approaches. We further show in Section 6.2 that our method can easily be used to supervise convolutional neural networks, when only relative orientation information is available.

## 6.1 Direct Parameter Optimization

We evaluate the convergence behaviour of our Iterative Modified Rodrigues Projective Averaging method, **MRP (Ours)** , described in Section 5, as well as the $SO(3)$ averaging method, described in Section 4. For the $SO(3)$ averaging method, we implement both the pure Riemannian optimization, **SO(3)**, as well as a method using a Projective Manifold Gradient Layer [29] to map the Riemannian gradient of the $SO(3)$ averaging loss, Equation 3a, to a Euclidean optimization in $\mathbb{R}^D$, where we test $D = 4, 6$, and 9, **4D PMG [29]**, **6D PMG [24]**, **9D PMG [25]**, respectively. Additionally, we evaluate direct quaternion optimization, described in Sections 4, **Quaternion**.

---

[2]The south pole in this case is described by the quaternion $-1 + 0i + 0j + 0k$

[3]We omit a constant factor for brevity, and integrate it into the learning rate, $\gamma$.

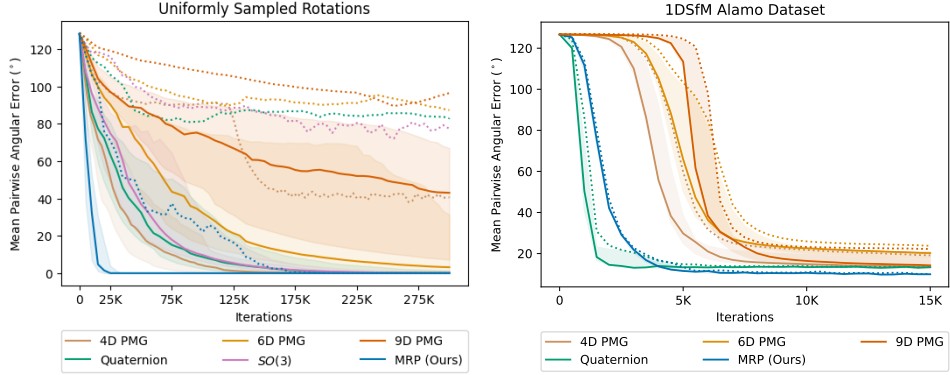

Figure 2: Relative rotation consensus with direct optimization of rotation parameters over 50 unique environments with 100 random generated orientations each (**left**) and Alamo 1DSfM [37] (**right**). Median average-pair-wise angular error ($^\circ$) between each estimated rotations is shown, with shaded region representing the first and third quartile for each method. The max average-pair-wise angular error for each algorithm at each iteration is shown as a dashed line.

| | Avg Pairwise Angular Error $< 5^\circ$ | | | Normalized AUC | | |
|---|---|---|---|---|---|---|
| Algorithm | Mean Steps | Max Steps | Min Steps | Mean | Max | Min |
| $SO(3)$ | 157.7K | Not Converged | 85.0K | 24.47 | 82.92 | 7.55 |
| 4D PMG [29] | 126.1K | Not Converged | 27.0K | 15.67 | 52.40 | 3.06 |
| 6D PMG [24] | 235.9K | Not Converged | 80.0K | 43.53 | 89.15 | 11.34 |
| 9D PMG [25] | 284.5K | Not Converged | 150.0K | 62.94 | 101.77 | 17.77 |
| Quaternion | 160.3K | Not Converged | 40.0K | 23.55 | 84.85 | 3.47 |
| **MRP (Ours)** | **37.5K** | **160.0K** | **15.0K** | **5.08** | **15.56** | **2.18** |

Table 1: Number of iteration steps until convergence and Normalized Area Under Curve (nAUC) over 50 unique environments of 100 randomly generated orientations. 300K optimization steps are taken for each experiment.

**Uniformly Sampled Rotations.** We test the performance of each algorithm when directly optimizing the rotation parameters of a set of size $N = 100$ with known relative rotations $R_i^j$, and local neighborhood structure. Ground truth and initial estimated rotations are both randomly sampled from a uniform distribution in $SO(3)$. Each rotation, $R_i$, has a neighborhood, $\mathcal{N}_i$, consisting of the closest $|\mathcal{N}_i| = 3$ rotations with respect to geodesic distance. The connectivity of this neighborhood graph is checked to ensure the graph contains only a single connected component. We test all algorithms over 50 sets of unique environments, each with $N = 100$ randomly generated orientations as described above. The estimated rotations are updated by each algorithm in batches of size 8, for 300K iterations.

As the goal of our algorithm is to improve the convergence properties of iterative averaging methods, we analyze each algorithm at various stages of optimization. We are particularly interested in the average number of update steps until the algorithm has converged, which we define as when the average angular error between all pairs of rotations is below $5^\circ$. As we can see in Figure 2, the Iterative Modified Rodrigues Projective Averaging method, **MRP (Ours)**, converges before the standard $SO(3)$ averaging method. On average, our method converged to within $5^\circ$ in 37K steps. The next best method, **4D PMG [29]**, which takes over three times as many iterations to converge to the same

| | % Avg Pairwise Angular Error $< 5^\circ$ | | | | | Final Error($^\circ$) | |
|---|---|---|---|---|---|---|---|
| Algorithm | 30K | 70K | 100K | 150K | 300K | Mean | Median |
| $SO(3)$ | 0% | 0% | 6% | 57% | 94% | 2.056 | 0.10 |
| 4D PMG [29] | 2% | 32% | 46% | 72% | 90% | 1.969 | 0.14 |
| 6D PMG [24] | 0% | 0% | 4% | 20% | 52% | 20.096 | 3.20 |
| 9D PMG [25] | 0% | 0% | 0% | 2% | 20% | 40.125 | 43.02 |
| Quaternion | 0% | 12% | 30% | 56% | 82% | 9.72 | 0.04 |
| **MRP (Ours)** | **66%** | **88%** | **96%** | **98%** | **100%** | **0.004** | **0.004** |

Table 2: Percentage of experiments converged and final angular errors over 50 unique environments of 100 randomly generated orientations. 300K optimization steps are taken for each experiment.

level of accuracy. Further, Table 1 shows that our method is the only one to converge across all environments within 300K iterations. For each method, we also compute the mean area under the pairwise error curve, with the number of steps normalized to between zero and one (nAUC), also shown in Table 1. We find that in the best, average, and worst case scenarios, our method has the best convergence behavior. To quantify convergence behavior, we also compute the percentage of trials that achieve average pairwise angular error below $5°$ at different stages of training, as shown on the left in Table 2. We find that at each stage of training, the Iterative Modified Rodrigues Projective Averaging, **MRP (Ours)**, training has a lower average pairwise error, shown in Table 2. Our method also converged far more often at each stage of training, also shown in Table 2.

**Structure from Motion Dataset.** To test our algorithms under natural noise conditions, we also evaluate our algorithm on the 1DSfM [37] structure from motions datasets. Each environment is tested with 5 random initializations and the estimated rotations are updated by each algorithm in batches of size 64, for 20K iterations. The results of a subset of the environments are shown in Table 3 and the remainder can be found in Appendix C. The noise characteristics of relative rotations in this dataset are similar to those found when capturing relative poses, but, unlike the Uniformly Sampled Rotations environments, the distribution of rotations does not fully cover the orientation space. As a result, all methods converge relatively quickly. Our algorithm outperforms the baselines in terms of accuracy. While the **Quaternion** optimization converges slightly faster, it consistently finds a lower accuracy configuration, resulting in a low nAUC, but higher relative and absolute accuracy. More details can be found in the Appendix C.

| | Mean Relative Error (°) | | Mean Absolute Error (°) | | Mean nAUC | |
|---|---|---|---|---|---|---|
| Algorithm | E. Island | Alamo | E. Island | Alamo | E. Island | Alamo |
| 4D PGM [29] | 11.94 | 15.00 | 7.34 | 9.94 | 25.60 | 47.20 |
| 6D PGM [24] | 11.26 | 18.84 | 6.90 | 13.09 | 27.77 | 58.04 |
| 9D PGM [25] | 10.22 | 16.32 | 6.32 | 11.43 | 29.31 | 60.14 |
| Quaternion | 11.58 | 13.40 | 7.23 | 8.93 | **16.01** | **22.57** |
| **MRP (Ours)** | **8.84** | **9.89** | **5.49** | **6.56** | 16.21 | 25.61 |
| IRLS-GM[17] | - | - | 3.04 | 3.64 | - | - |
| IRLS-$\ell_{\frac{1}{2}}$ [18] | - | - | 2.71 | 3.67 | - | - |
| MLP[19] | - | - | 2.61 | 3.44 | - | - |

Table 3: **Rotation Averaging Results on 1DSfM [37] dataset.** Results before the double lines are comparisons of local method after 20K iterations. Results under the double lines are obtained from global methods which are incompatible with SGD training.

## 6.2 Neural Network Optimization

To show that the Iterative Modified Rodrigues Projective Averaging method, **MRP (Ours)**, can be used to learn orientation using neural networks by optimizing the parameters of a simple CNN, specifically a ResNet18 [38], we follow the procedure as in Section 6.1 with some minor changes. Instead of operating directly on a set of rotation parameters, we learn a function $\hat{\psi}_i = f(I_i)$ from rendered images of the YCB drill [1] model rendered at each of 100 random orientations $R_i$. We continue to only supervise each method described in Section 6.1 using the relative rotations between each image.

| Algorithm | Mean Error (°) | Median Error (°) | $5°$ Acc (%) |
|---|---|---|---|
| 4D PMG [29] | 123.84 | 123.96 | 0 |
| Quaternion | 28.83 | 21.74 | 50 |
| **MRP (Ours)** | **3.71** | **3.73** | **100** |
| Oracle | 1.58 | 1.56 | 100 |

Table 4: **Final results for image based rotation estimation.** Final mean and median angular error (°) and percentage of seeds below $5°$ after 10K steps over 8 unique environments of 100 random rotations.

We compare the best performing methods, and, as a lower bound, we also train an oracle network, **Oracle**, with the ground truth rotations, $R_i$ and cosine quaternion loss. We use a batch size of 32 and the Adam [39] optimizer with a learning rate of $10^{-4}$ for all experiments. All methods are trained for a maximum of 10K steps, over 8 environments, each with 100 images of randomly generated rotations. We report final mean and median pairwise angular error, and the percentage

| | Final Test Angular Pairwise Error | | | | | Final Test Angular Pairwise Error | | |
|---|---|---|---|---|---|---|---|---|
| Algorithm | Mean (°) | Max (°) | Min (°) | | Algorithm | Mean (°) | Max (°) | Min (°) |
| 4D PMG [29] | $17.39 \pm 1.14$ | 19.42 | 16.07 | | 4D PMG [29] | $34.57 \pm 2.21$ | 38.13 | 31.90 |
| 6D PMG [29, 24] | $15.20 \pm 0.77$ | 16.43 | 14.44 | | 6D PMG [29, 24] | $31.58 \pm 2.24$ | 35.66 | **28.42** |
| 9D PMG [29, 25] | $14.61 \pm 0.50$ | 15.66 | 14.18 | | 9D PMG [29, 25] | $31.80 \pm 1.52$ | 34.87 | 29.96 |
| 10D PMG [29, 26] | $19.28 \pm 7.58$ | 37.76 | 15.03 | | 10D PMG [29, 26] | $32.23 \pm 2.10$ | 36.98 | 29.87 |
| Quaternion | $16.52 \pm 4.12$ | 26.57 | 14.38 | | Quaternion | $31.92 \pm 1.00$ | 33.61 | 30.61 |
| **MRP (Ours)** | $\mathbf{13.63 \pm 0.78}$ | **15.08** | **12.62** | | **MRP (Ours)** | $\mathbf{29.46 \pm 0.66}$ | **30.74** | 28.62 |
| | (a) | | | | | (b) | | |

Table 7: **3D Object Pose Estimation via Relative Supervision on Pascal3D+ test images for *sofa*, *bicycle*.** Final mean test angular pairwise error on Pascal3D+ *sofa* (a), *bicycle* (b) images after 80K training iterations, over 8 random seeds.

of runs converged below $5°$ pairwise angular error as $5°$ Acc. We find that **MRP (Ours)** is able to converge to a rotational frame consistent with the relative rotations used for supervision relatively quickly, with a significantly lower average-pairwise-error than all other relative methods, shown in Table 4.

We perform experiments for generalization to unseen poses and find that a curriculum is required (see Appendix D for details). For the generalization experiments, we found that **MRP (Ours)** achieves a mean pairwise angular error or $5.19°$, **Quaternion** achieves $12.41°$, and **4D PMG [29]** never converged, with final error of $125.09°$.

## 6.3 3D Object Rotation Estimation via Relative Supervision from Pascal3D+ Images

**Experimental Setup.** Pascal3D+ [40] is a standard benchmark for categorical 6D object pose estimation from real images. Following [29, 25], we discard occluded or truncated objects and augment with rendered images from [23]. We report 3D object pose estimation results on two object categories: *sofa* and *bicycle*. We compare our method **MRP** with five baselines: **Quaternion**, **4D PMG** [29], **6D PMG** [24], **9D PMG** [25] and **10D PMG** [26], all of which use ResNet18 [38] backbone to predict the object representation. Each model is supervised using the geodesic error between the relative orientation of the predicted absolute orientations, and the relative orientation between the ground truth absolute orientations, for each image pair. We use the same batch size of 20 as in [25, 29], and use Adam [39] with learning rate of $10^{-4}$.

**Result Analysis.** Results for *sofa* are showed in Table 7(a); *bicycle* are showed in Table 7(b). For *sofa* category, we find that after 80K training iterations, **MRP (Ours)** achieves a mean angular pairwise error of $13.63°$ on the test set, outperforms all other baselines. **10D PMG** achieves the worst error out of all methods, with final angular pairwise error of $19.28°$. For *bicycle* category, we find that after 80K training iterations, **MRP (Ours)** achieves a mean angular pairwise error of $29.46°$ on the test set, outperforms all other baselines. In both the *sofa* and the *bicycle* category, we find that **MRP (Ours)** has the fastest convergence speed, in addition to achieving the lowest test angular error. More details can be found in Appendix E.

# 7 Conclusion and Limitations

In this paper, we show that through the use of Modified Rodrigues Parameters, we are able to open the closed manifold of $SO(3)$, improving the convergence behavior of the rotation averaging problem. Additionally, we show that our method, Iterative Modified Rodrigues Projective Averaging, is able to outperform the naive application of relative-orientation supervision in both direct parameter optimization and image-based rotations estimation from neural networks. While this parameterization is valuable for learning rotations through relative supervision, it is not without limitations. One of the primary ones is the need for a curriculum to generalizing to unseen relative rotations. Without this, our experiment show that all representations fall into the local optima of predicting a constant orientation. Additionally, in the generalization experiments, we were only able to achieve a final error of 5 degrees, which may not be accurate enough for many fine motor tasks. We hope our method allows more systems to convert the relative supervision of relative methods, like ICP, to consistent and accurate absolute poses.

# 8 Acknowledgements

We thank Prof Zachary Manchester in particulate for pointing us to Modified Rodriguez Parameters. Additionally, we thank Prof Frank Dellaert, Mark Gillespie, Prof Richard Hartley, Valentin Peretroukhin, Prof David Rosen, Prof Denis Zorin, and Prof Keenan Crane for taking time to meet with us and providing enormously helpful feedback and advice. This work is supported by the National Science Foundation under Grant No. IIS-1849154 and by LG Electronics.

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
