# OpenReview forum: "Deep Projective Rotation Estimation through Relative Supervision"
_robot-learning.org/CoRL/2022/Conference — CoRL 2022 Poster_

### Official Review · Reviewer_2N3Q · 2022-07-04

**Originality:** Very Good
**Technical Quality:** Poor
**Clarity Of Presentation:** Good
**Impact:** 2

**Recommendation:**

Weak Accept: I recommend accepting the paper, but will not argue for my recommendation if the majority of other reviewers have a different opinion.

**Summary:**

The paper uses modified Rodrigues parameters for estimating absolute rotations from relative rotations, which can be used within "self-supervised" training of neural networks. In this training regime, it is assumed that relative orientations between neighboring poses are given, for example, computed from a local alignment method like ICP, and that a neural network should be trained by using the relative poses as supervision. The paper shows that the proposed scheme can obtain better results than baseline methods for synthetic data and 1DSfM data when the rotations are optimized directly. Additionally, the paper shows a synthetic example where the rotations are not inferred directly, but a neural network is trained to infer the rotations from rendered images.

**Issues:**

The paper would be improved significantly if the authors can provide extensive evaluations on real-world learning tasks wrt. established SOTA methods on established benchmarks. However, this might be too much for a rebuttal and should be done when potentially re-submitting the paper.

**Quality Of The Limitations Section:**

Limitations are not well addressed

**Reviewer Expertise:**

3: The reviewer is fairly confident that the evaluation is correct

**Robotics Focus:**

Relevant but unlikely to deploy to hardware in near future

**Strengths And Weaknesses:**

**Strengths:**

- The paper proposes an interesting method for rotation learning that is based on modified Rodrigues parameters. It nicely combines current deep learning with traditional works on rotation parametrization.
- The paper is overall well-written and the content is well-presented.


**Weaknesses**:

- The paper does not show results for an actual learning problem. The shown examples either have no learning (Sec. 6.1) or only synthetic data with overfitting, i.e. no learning as well (Sec 6.2).
- The paper does not show results on recognized benchmarks. Sec. 6.1 uses the 1DSfM dataset but does not compare with global methods, which are SOTA there. The other experiments use data generated by the authors.

**Summary Of Recommendation:**

The paper proposes a seemingly novel and interesting idea, however, an evaluation on actual learning tasks and real-world data is missing. This makes it impossible to judge the benefit of the proposed method for robot learning. If the paper would show extensive experiments on real-world learning tasks I would be willing to increase my score.

---

> ### Author Response · Authors · 2022-08-27
> **Response to Reviewer 2N3Q**
>
> **Comment:**
>
> Thank you for the valuable feedback and suggestions! The raised questions are addressed as follows.
>
> We color the revised text and tables in magenta in the newly revised paper and supplement attached both as a merged pdf and as a zip folder.
>
> **Q: “The paper does not show results for an actual learning problem. The shown examples either have no learning (Sec. 6.1) or only synthetic data with overfitting, i.e. no learning as well (Sec 6.2).”**
>
> **A:** Thank you for this question! In our paper, we first demonstrated that, even with such a simple setup that does not involve learning, many of the previous methods fail to converge when trained using only relative supervision.  In contrast, our method converges well in such a case.
>
> Additionally, in our original submission, we had performed an experiment that showed learning with generalization to new poses, described at the end of Section  6.2, with further details in Supplement Section C.
>
> Furthermore, we have added new results on learning on *Pascal3D+* image dataset in Supplement Section D and *ModelNet40* point cloud dataset in Supplement Section E, which demonstrate generalization to new poses and new model instances. We report 3D object orientation estimation via relative orientation supervision from ModelNet40 point clouds of the *airplane* categories and from Pascal3D+ images for the *sofa* and *bicycle* categories. We compare our method MRP with 5 baselines: **Quaternion**, **4D PMG** [32], **6D PMG** [25, 32], **9D PMG** [26, 32], **10D PMG** [27, 32]. We see similar patterns as seen in our previous experiments, where MRP (ours) achieves better final pairwise angular error on the Pascal3D+ test set on each category (shown in Figure S.3, Table S6, Figure S.4, Table S7 in Supplement Section D) and on the the ModelNet40 point cloud dataset (Figure S.5, Table S8 in Supplement Section E). We show that our method outperforms previous approaches in terms of both final test pairwise angular error as well as convergence speed on these established benchmarks as well.
>
> **Q: “The paper does not show results on recognized benchmarks. Sec. 6.1 uses the 1DSfM dataset but does not compare with global methods, which are SOTA there. The other experiments use data generated by the authors.”**
>
> **A:** We have added the results from global methods to Table 3 in the main paper and to Table 1 & 2 in the Supplement Section B , with the caveat that these methods can not be easily used to supervise neural networks, as they 1) rely on semidefinite programing; 2) are global methods and aren’t designed for local batch-wise optimization. We have additionally added learning results on more recognized benchmarks: Pascal3D+ images in Supplement Section D and ModelNet40 point clouds in Supplement Section E and show that our method outperforms previous approaches on these established benchmarks.
>
> *We greatly appreciate your feedback, and please let us know if these changes have addressed your concerns and if there are other questions you may have!*
>
>
> **Zip File:**
>
> /attachment/3ee20dda1f94ea9c636cc67ea66a4f9c7b91b910.zip

---

### Official Review · Reviewer_WRCQ · 2022-07-22

**Originality:** Fair
**Technical Quality:** Good
**Clarity Of Presentation:** Good
**Impact:** 3

**Recommendation:**

Weak Accept: I recommend accepting the paper, but will not argue for my recommendation if the majority of other reviewers have a different opinion.

**Summary:**

This paper proposed a self-supervised rotation estimation method for object or camera pose estimation. Major contributions are as follows:

(1) Author proposed to use modified Rodrigues parameters to stereographically project the closed manifold of $SO(3)$ to the open manifold of $R^3$, which can speed up the optimization and avoid the local optima.

(2) The proposed optimization could be applied to the Stochastic Gradient Descent of the training for neural networks.

**Issues:**

Authors should clearly state their novelty and contribution against the prior works.

**Quality Of The Limitations Section:**

Limitations are addressed clearly

**Reviewer Expertise:**

3: The reviewer is fairly confident that the evaluation is correct

**Robotics Focus:**

Highly relevant to robotics but no hardware experiments

**Strengths And Weaknesses:**

**Strengths:**
* According to the experiment section, the proposed Iterative Modified Rodrigues Averaging could improve the convergence behavior of the rotation averaging problem.
**Weaknesses:**
* The novelty statement is unclear. Line 82, "the unique problems found in the rotation averaging problem" need more detatils.

**Summary Of Recommendation:**

The experiment section looks convincing. My major concern is the novelty and contribution compared to [24] and [25] are unclear.

Post-rebuttal:
The reply from the author partially resolves my concern, I would keep my rating as "weak accept", I think it deserves a poster presentation.

---

> ### Author Response · Authors · 2022-08-27
> **Response to Reviewer WRCQ**
>
> **Comment:**
>
> Thank you for the valuable feedback and suggestions! The raised questions are addressed as follows.
>
> We color the revised text and tables in magenta in the newly revised paper and supplement attached both as a merged pdf and as a zip folder.
>
> **Q: “My major concern is the novelty and contribution compared to [24] and [25] are unclear.”**
>
> **A:** While we are not the first to propose using MRP introduced by [25] for computer vision, we are the first to analyze its use with respect to 1) relative pose supervision, 2) stochastic gradient descent and 3) rotation estimation via neural nets. All experiments in [24] are supervised with absolute pose and used direct parameter optimization as opposed to relative pose supervision with neural net optimization.
>
> The primary focus of this work is to examine the unique problems that arise when using relative supervision for orientation estimation with stochastic gradient descent and neural networks.  We show that previously used rotational losses often fail to converge in such a setting, whereas our approach leads to significantly improved performance in such settings.
>
> Additionally, our analysis of relative supervision indicates that curriculum learning is usually necessary for pose generalization when training with relative supervision with neural networks on orientation domains that fully cover the space of SO(3), as seen by experiments in Sec 6.1, and the additional experiments on 3D object orientation estimation with relative supervision from ModelNet40 point cloud dataset added in Supplement Section E.
>
> *We greatly appreciate your feedback, and please let us know if these changes have addressed your concerns and if there are other questions you may have!*
>
>
> **Zip File:**
>
> /attachment/bd63c2e4352221d7b3fdecf2358d0f03ff367079.zip

---

### Official Review · Reviewer_cxjP · 2022-07-30

**Originality:** Fair
**Technical Quality:** Fair
**Clarity Of Presentation:** Good
**Impact:** 4

**Recommendation:**

Strong Accept: I recommend accepting the paper and will argue for my recommendation even if other reviewers hold a different opinion.

**Summary:**

The authors propose the use of Modified Rodrigues Parameters (MRP), which maps Rotations in $SO(3)$ to vectors in $\mathbb{R}^3$, in an incremental manner so that it is suited for learning with gradient descent. They look at self-supervision using relative rotations, where given a set of inputs the relative rotations w.r.t. a subset of the inputs is known, such as by using ICP or camera tracking. This trick of optimizing in the space of MRP circumvents issues that come with $SO(3)$ as it is a closed manifold. The intuition behind the method is shown well in illustrative toy examples making it easy to understand. The effectiveness and efficiency of the approach are shown on two datasets, a toy dataset of uniformly sampled rotations and the 1DSfM dataset on which direct rotation parameter estimates are optimised. Good performance is also shown in an initial experiment of training neural networks on images for relative object pose estimation for synthetic images of a single object.


**Issues:**

- While the use of optimisation for predicting relative transforms is an old one in computer vision, there have been approaches that explore the use of such optimization (like non-linear least squares) in a differentiable way within neural network frameworks trained with gradient descent (BA-Net [1], or Deep Direct VO [2] who use LM as part of the Network optimisation). I would imagine that such an approach could work in such a case of rotation averaging when considering a minibatch of samples with their neighbourhood, given that such optimisation approaches are known for handling not just $SO(3)$ manifolds but $SE(3)$ as well.

[1] [Tang, C. and Tan, P., 2018, September. BA-Net: Dense Bundle Adjustment Networks. In International Conference on Learning Representations (ICLR).](https://arxiv.org/abs/1806.04807)

[2] [Wang, C., Buenaposada, J.M., Zhu, R. and Lucey, S., 2018. Learning depth from monocular videos using direct methods. In IEEE Conference on Computer Vision and Pattern Recognition (CVPR).](https://openaccess.thecvf.com/content_cvpr_2018/papers/Wang_Learning_Depth_From_CVPR_2018_paper.pdf)

- MRP is not a contribution of the paper but is something behind the base of the approach, maybe calling the approach iMRPAvg or something to emphasise the fact that the proposed approach is an iterative Version encapsulating MRP would be better.
- ~~In Table 4 what is Acc $5^\circ$? Is it the same as "Avg Pairwise Angular Error < $5^\circ$ " in Table 1? This needs to be clarified in the text.~~

(Post-Rebuttal Updates)
- As mentioned above, Section F doesn't supplement the proposed approach but rather shows the advantage of MRP. It would have been better to show the absolute orientations of using the Iterative MRP approach proposed by the authors.
- The formatting of Tables S1 and S2 in the supplementary material need to be fixed.
- The XYZ axes that are overlayed on the images (Figures 3, S6-S8) can be made thicker as they are very thin to see properly. Additionally, Figures S8 and S7 seem to have an overlay on them which makes it difficult to see the images clearly.
- I would suggest the authors to export the graphs in vector graphics (SVG) or PDF formats in order to prevent pixelation of the images.

**Quality Of The Limitations Section:**

Additional details required

**Reviewer Expertise:**

4: The reviewer is confident but not absolutely certain that the evaluation is correct

**Robotics Focus:**

Highly relevant to robotics but no hardware experiments

**Strengths And Weaknesses:**

### Strengths
- One of the first works exploring the use of MRP for Gradient Descent approaches rather than global optimisation.
- show faster convergence when training direct rotation parameter estimates and neural networks and also overall low error against baseline methods ( less than $5^\circ$ for pairwise relative poses)
- provide good illustrative toy examples (with visualisations on their website) allowing the reader/viewer to easily understand the problems with optimising in $SO(3)$ and how these are circumvented in $\mathbb{R}^3$ and how their iterative approach enables faster convergence.
- The paper is nicely written and has a good progressive flow to understand the problem that is looked at and the approach presented by the authors.

(Post-Rebuttal Updates)
- The additional results shown in the supplementary material on both synthetic object poses and on real world datasets of Objects (Pascal3D+) as well outdoor scenes (1DSfM) show the efficacy of the proposed approach compared to other methods using relative supervision. The addition of visual results showing the predicted relative pose predictions helped.

### Weaknesses
- ~~Since the main focus of the work is on deriving an iterative version of MRP averaging, such that it is better suited to work with Neural Network of Gradient Descent approaches, the use of only simulated images of a single object is a very simplistic evaluation for such a claim. While the relative pose estimation with neural networks shows good initial results, its ability to generalise cannot be fully ascertained using only simulated images of a single object.~~
 - ~~The paper (and supplementary) is lacking in visual qualitative results that can better help a reader to understand the results. Visualising some results for example the intermediate results of training the neural networks or the alignments of the 1DSfM data could help one better understand the approach.~~

(Post-Rebuttal Updates)
- I did not understand the need for Section F in the appendix. This essentially shows the advantage of just MRP and not the proposed approach. I would imagine that the use of Iterative MRP as a representation for absolute orientation could also be useful for the results.

**Summary Of Recommendation:**

The authors propose the use of MRP for training gradient descent approaches. While the idea is simplistic, they are the first to look at this novel use of MRP in a relevant use case in robot learning. The illustration of existing issues in optimising in $SO(3)$ manifolds is well illustrated, motivating the idea of using MRP to transform the space of the loss landscape. The idea, although incremental, is novel enough to be worth considering as part of the conference. However, the simplistic nature of the evaluations, especially in the context of neural networks, prevents me from completely advocating for a strong acceptance, especially when the goal is to train neural networks well for predicting rotations. ~~The paper could benefit from a stronger set of experiments on a more diverse dataset rather than on a single object (from an already diverse dataset), especially when considering training neural networks. For example, by training existing approaches like those presented in the Related Works section using the MRP representation, I would imagine that other than just relative rotations, even learning absolute rotations could benefit by using MRP representations during training. While I would like to give more than just a weak accept, since the only next level is a strong accept without anything in between, my final recommendation is a weak accept.~~

### Post-Rebuttal Updates
The updated results show a good efficacy of the proposed approach on a range of different datasets when compared with different relative rotation estimation approaches. I did not fully understand Section F in the supplementary as it doesn't showcase the usefulness of the proposed approach but rather shows the power of using MRP. Nevertheless, the iterative extension of MRP making it suitable for use in gradient descent settings and the promising results warrant an acceptance.

---

> ### Author Response · Authors · 2022-08-27
> **Response to Reviewer cxjP**
>
> **Comment:**
>
> Thank you for the valuable feedback and suggestions! The raised questions are addressed as follows.
>
> We color the revised text and tables in magenta in the newly revised paper and supplement attached both as a merged pdf and as a zip folder.
>
> **Q: “Since the main focus of the work is on deriving an iterative version of MRP averaging, such that it is better suited to work with Neural Network of Gradient Descent approaches, the use of only simulated images of a single object is a very simplistic evaluation for such a claim. While the relative pose estimation with neural networks shows good initial results, its ability to generalize cannot be fully ascertained using only simulated images of a single object."**
>
> **A:** We agree with the reviewer that this is a simplistic evaluation, and would like to clarify that our goal was to use the simplest possible environment to show that even in these simple cases, relative supervision can fail when applied using baseline rotation parametrizations. We would like to clarify that we have evaluated on the 1DSfM dataset [40], which has been used in previous works [18, 19, 20]. However, we agree that evaluating our method on more benchmarks would be helpful to strengthen the claim; thus we have subsequently tested our method on **ModelNet40** point cloud dataset and **Pascal3D+** image datasets. The Pascal3D+ results are added to Supplement Section D, ModelNet40 results are added in Supplement Section E.
>
> We follow a similar experiment setup as [32] for both experiments on ModelNet40 and Pascal3D+ datasets. We report 3D object orientation estimation via relative orientation supervision from ModelNet40 point clouds of the *airplane* categories and from Pascal3D+ images for the *sofa* and *bicycle* categories. We compare our method MRP with 5 baselines: **Quaternion**, **4D PMG** [32], **6D PMG** [25, 32], **9D PMG** [26, 32], **10D PMG** [27, 32]. We see similar patterns as seen in our previous experiments, where MRP (ours) achieves better final pairwise angular error on the Pascal3D+ test set on each category (shown in Figure S.3, Table S6, Figure S.4, Table S7 in Supplement Section D) and on the the ModelNet40 point cloud dataset (Figure S.5, Table S8 in Supplement Section E). And we show that our method outperforms previous approaches in terms of both final test pairwise angular error as well as convergence speed on these established benchmarks as well.
>
> **Q: "The paper (and supplementary) is lacking in visual qualitative results that can better help a reader to understand the results. Visualising some results for example the intermediate results of training the neural networks or the alignments of the 1DSfM data could help one better understand the approach."**
>
> **A:** Thanks for the feedback and suggestions! We have added qualitative object orientation visual results on both the Pascal3D+ experiments and the YCB drill experiment to Supplement Section G, in Figure S.6, S.7 and S.8. We hope this help the readers understand the prediction results better in a qualitative manner.
>
> **Q: “In Table 4 what is Acc $5^{\circ}$? Is it the same as "Avg Pairwise Angular Error $< 5^{\circ}$ " in Table 1? This needs to be clarified in the text.”**
>
> **A:** We apologize for the confusion caused. *Acc $5^{\circ}$* refers to the percentage of experiment runs that converge below pairwise angular error of $5^{\circ}$, which is the same as the “% Avg Pairwise Angular Error $< 5^{\circ}$” in Table 2 in the main text. We have updated the caption in Table 4 and Sec 6.2  in the paper to clarify this.
>
> *We greatly appreciate your feedback, and please let us know if these changes have addressed your concerns and if there are other questions you may have!*
>
>
> **Zip File:**
>
> /attachment/a25a82698ae7be66ad08fdfe13da0b46d1347298.zip

---

### Official Review · Reviewer_ziM5 · 2022-07-31

**Originality:** Very Good
**Technical Quality:** Good
**Clarity Of Presentation:** Very Good
**Impact:** 3

**Recommendation:**

Weak Accept: I recommend accepting the paper, but will not argue for my recommendation if the majority of other reviewers have a different opinion.

**Summary:**

The paper analyzes the problem of learning orientation via self-supervised learning, especially via relative orientation data. Specifically, the paper proposes to use a new representation via Modified Rodrigues Parameters (MRP) and claim that this provides more stable learning convergence when compared to previous methods which use different rotational spaces like SO(3).

The paper then analyzes MRP experimentally on experiments including uniformly sampled rotations, the 1DSfM dataset, and an image-based neural network optimization problem. The results show that MRP improves results upon the baselines compared to previous methods.

**Issues:**

Some summary of issues that could be addressed from above:

- (Fig 2 left)  Are all methods fully converged? Would running more steps improve some methods?
- A comparison on some benchmark used in prior work, e.g. [28: modelnet40 point clouds / modelnet10 images] [21: ​​CMU Motion Capture Database] [22: pascal3D+]
- How does MRP and the other relative rotation methods compare to using direct supervised learning? How far does MRP bridge the gap to supervised learning techniques? e.g. Should practitioners considering supervised methods here instead use MRP?
- The neural network optimization benchmark is evaluated on only one object, the YCB drill. While the method performs much better than prior work for this model, it would’ve been more convincing that this method is very compatible with neural network based optimization if an entire set of objects were tested. This may help answer some other questions, e.g.
  - Would this method work better for image-based optimization if the object has some symmetries?
  - How may a practitioner envision using this algorithm in a larger end-to-end neural network based learning problem? How practical would unseen relative rotations or learning curriculums be in this end-to-end case?


**Quality Of The Limitations Section:**

Limitations are addressed clearly

**Reviewer Expertise:**

3: The reviewer is fairly confident that the evaluation is correct

**Robotics Focus:**

Highly relevant to robotics but no hardware experiments

**Strengths And Weaknesses:**

Strengths
+ The authors are clear in the theoretical explanations of MRP and in the visuals and examples
+ The authors show clear benefit of using MRP for estimating orientation via relative rotations compared to previous methods, on the evaluated datasets
+ The authors run experiments across multiple datasets and include an end-to-end example using a deep neural network
+ The experiments show that MRP converges faster than previous methods, supporting the authors’ claims
+ The method is able to be used as a relatively straightforward, drop-in replacement for other rotation representations for the self-supervised relative rotation problem, and this ease may expand its impact

Weaknesses

- The authors show that MRP outperforms prior methods in evaluation results, but some charts like in Fig. 2. (left) show that some other methods may not have fully converged. It would be good to understand whether MRP would converge to a better performance compared to other methods, after they have all converged.
- The prior work compared against [21,22,28] use datasets/benchmarks, but none are covered in this paper, e.g. [28: modelnet40 point clouds / modelnet10 images] [21: ​​CMU Motion Capture Database] [22: pascal3D+] and instead novel benchmark sets are used
- How does MRP and the other relative rotation methods compare to using direct supervised learning? How far does MRP bridge the gap to supervised learning techniques? e.g. Should practitioners considering supervised methods here instead use MRP?
- The neural network optimization benchmark is evaluated on only one object, the YCB drill. While the method performs much better than prior work for this model, it would’ve been more convincing that this method is very compatible with neural network based optimization if an entire set of objects were tested. This may help answer some other questions, e.g.
  - Would this method work better for image-based optimization if the object has some symmetries?
  - How may a practitioner envision using this algorithm in a larger end-to-end neural network based learning problem? How practical would unseen relative rotations or learning curriculums be in this end-to-end case?


**Summary Of Recommendation:**

Taken as a whole, this work makes a clear case for use of MRP via theoretical explanations and in isolated experimental results for use in relative orientation self-supervised learning. However, the larger impact of the work is tied to the use and feasibility of self-supervised approaches for orientation learning compared to supervised ones, which is not examined via experiments. This limits the conclusion of the work - it is not clear whether practitioners should switch to using MRP, which requires using self-supervised methods as a whole. Noted, the authors make clear mitigations or concerns of self-supervised methods like curriculum requirements.

Limiting scope to improvement of rotation representation for self-supervised methods, there also lacks direct comparison to benchmarks used in prior work and a comprehensive picture of neural-network based performance. Thus, while improvement looks promising given the presented results, it would be hard to draw a comprehensive, apples-to-apples conclusion.


Edit: post review, the authors have sufficiently addressed my concerns and it is clear that the work is promising for usage/consideration by practitioners to improve neural network rotation prediction.

---

> ### Author Response · Authors · 2022-08-27
> **Response to Reviewer ziM5 (1/2)**
>
> **Comment:**
>
> Thank you for the valuable feedback and suggestions! The raised questions are addressed as follows.
>
> We color the revised text and tables in magenta in the newly revised paper and supplement attached both as a merged pdf and as a zip folder.
>
> **Q: “The authors show that MRP outperforms prior methods in evaluation results, but some charts like in Fig. 2. (left) show that some other methods may not have fully converged. It would be good to understand whether MRP would converge to a better performance compared to other methods, after they have all converged.”**
>
> **A:** Thank you for the feedback. We have updated Figure 2 in the paper to show evaluation results after 300K optimization steps. As illustrated by the updated Figure 2, we find that even the worst case run of MRP converges after 175K iterations (shown by the dashed blue line), whereas the worst case run of all other parameterizations (shown by the other dashed lines) have not yet converged even after 300K optimization steps.
>
> **Q: “The prior work compared against [21,22,28] use datasets/benchmarks, but none are covered in this paper, e.g. [28: modelnet40 point clouds / modelnet10 images] [21: CMU Motion Capture Database] [22: pascal3D+] and instead novel benchmark sets are used”**
>
> **A:** First, we would like to clarify that we have evaluated on the 1DSfM dataset [40], which has been used in previous works [18, 19, 20].
>
> Following the reviewer suggestion, we have also tested our method on **ModelNet40**  point cloud dataset and **Pascal3D+** image datasets; the Pascal3D+ results added to Supplement Section D, ModelNet40 results are added in Supplement Section E. We follow a similar experiment setup as [32] for both experiments on ModelNet40 and Pascal3D+ datasets. We report 3D object orientation estimation via relative orientation supervision from ModelNet40 point clouds of the *airplane* categories and from Pascal3D+ images for the *sofa* and *bicycle* categories. We compare our method MRP with 5 baselines: **Quaternion**, **4D PMG** [32], **6D PMG** [25, 32], **9D PMG** [26, 32], **10D PMG** [27, 32]. We see similar patterns as seen in our previous experiments, where MRP (ours) achieves better final pairwise angular error on the Pascal3D+ test set on each category (shown in Figure S.3, Table S6, Figure S.4, Table S7 in Supplement Section D) and on the the ModelNet40 point cloud dataset (Figure S.5, Table S8 in Supplement Section E). We show that our method outperforms previous approaches in terms of both final test pairwise angular error as well as convergence speed on these established benchmarks as well.
>
> **Q: “How does MRP and the other relative rotation methods compare to using direct supervised learning? How far does MRP bridge the gap to supervised learning techniques? e.g. Should practitioners considering supervised methods here instead use MRP?”**
>
> **A:** This is a good question. To clarify, our method is designed to help when direct supervision is not possible because the user does not have access to ground-truth rotation annotations. In such a case, we demonstrate that many rotation representations fail to converge from such relative supervision when trained with gradient descent, whereas our approach converges consistently to a better final performance than the other methods.
>
> Still, following the reviewer suggestion, we performed additional experiments to explore how different orientation representations perform if direct rotation supervision is available.  We have added experimental results in Table S9 in Supplement Section F. We find that when using direct supervision, MRP performs similar to the quaternion representation, achieving 1.81 degree mean angular error (MRP) compared to 1.58 degree (quaternion). Both MRP and quaternion representations achieve 87.5% convergence below 2 degree angular error. Thus in the case of direct pose supervision, MRP may not be the best choice of rotation representation; using an open manifold such as in MRP is beneficial only in the case of relative pose supervision.
>
> **Q: “Would this method work better for image-based optimization if the object has some symmetries?”**
>
> **A:** We thank the reviewer for this question. At the moment, we don’t believe that our method should be any more effective in handling symmetries than most other rotation representations.
>
>
> **Zip File:**
>
> /attachment/8d615ed1a5e0b185a9c35a7cc4be1c2f20fd2a25.zip

---

> > ### Author Response · Authors · 2022-08-27
> > **Response to Reviewer ziM5 (2/2)**
> >
> > **Q: “How may a practitioner envision using this algorithm in a larger end-to-end neural network based learning problem? How practical would unseen relative rotations or learning curriculums be in this end-to-end case?”**
> >
> > **A:** Thank you for the great question! One practical approach could be as follows: given a video, a curriculum could be established based on temporal proximity in the video. One can choose any initial frame as the starting point for the video and train over a curriculum of increasing temporal distance to neighboring frames until the entire video has been trained on. We have added such a discussion to the end of Curriculum Discussion in Supplement Section C.
> >
> > *We greatly appreciate your feedback, and please let us know if these changes have addressed your concerns and if there are other questions you may have!*

---

### Author Response · Authors · 2022-08-27
**Revised paper + supplement**

**Comment:**

The modified section are colored in magenta in the updated revised paper and supplement!

**Zip File:**

/attachment/10414f0df23dddbd24899bd37c6b4d75714944a2.zip

---

### Meta-Review · Area_Chair_mDG5 · 2022-08-13

**Recommendation:** Accept (Poster)
**Confidence:** 4

**Metareview:**

The paper proposes a self-supervised learning approach to solve pose estimation. The method uses a new representation via Modified Rodrigues Parameters (MRP) to solve the problem of learning orientation via relative orientation data. The method is evaluated against uniformly sampled rotations, the 1DSfM dataset, and an image-based neural network optimization problem.

While most of the reviewers agree that the paper is well written especially with good illustrative examples, and the benefit of using MRP is clear in terms of convergence, they brought up several aspects of improvement for results:

(1) Lacks direct comparison to benchmarks used in prior work.

(2) Generalization is questionable as it only uses simulated images with single object.

(3) Evaluation on actual learning tasks and real world data will make the paper stronger.

During the reviewer discussion, most reviewers acknowledged that their major concerns are addressed in the rebuttal. Reviewers agree the additional results in the revised paper show their method's usefulness with unseen objects. For the above reasons, we recommend accepting the paper as poster.

---

> ### Author Response · Authors · 2022-08-27
> **Response to Area Chair**
>
> **Comment:**
>
> Thank you for the valuable feedback and suggestions! The raised questions are addressed as follows.
>
> Note: We color the revised text and tables in magenta in the newly revised paper and supplement attached both as a merged pdf and as a zip folder.
>
> **Q: “(1) Lacks direct comparison to benchmarks used in prior work.”**
>
> **A:** We have added the results from global methods to Table 3 in the main paper and to Table S1 & S2 in the Supplement Section B, with the caveat that these methods can not be easily used to supervise neural networks, as they 1) rely on semidefinite programing; 2) are global methods and aren’t designed for local batch-wise optimization. We have additionally added learning results on more recognized benchmarks: Pascal3D+ images in Supplement Section D and ModelNet40 point clouds in Supplement Section E and show that our method outperforms previous approaches on these established benchmarks.
>
> **Q: “ (2) Generalization is questionable as it only uses simulated images with single object. (3) Evaluation on actual learning tasks and real world data will make the paper stronger.”**
>
> **A:** Following the reviewer suggestion, we have also tested our method on the **ModelNet40** point cloud dataset and **Pascal3D+** image datasets; the Pascal3D+ results have been added to Supplement Section D, and ModelNet40 results have been added in Supplement Section E. We follow a similar experiment setup as [32] for both experiments. We report 3D object orientation estimation via relative orientation supervision from ModelNet40 point clouds of the *airplane* categories and from Pascal3D+ images for the *sofa* and *bicycle* categories. We compare our method MRP with 5 baselines: **Quaternion**, **4D PMG** [32], **6D PMG** [25, 32], **9D PMG** [26, 32], **10D PMG** [27, 32]. We see similar patterns as seen in our previous experiments, where MRP (ours) achieves better final pairwise angular error on the Pascal3D+ test set on each category (shown in Figure S.3, Table S6, Figure S.4, Table S7 in Supplement Section D) and on the the ModelNet40 point cloud dataset (Figure S.5, Table S8 in Supplement Section E). We show that our method outperforms previous approaches in terms of both final test pairwise angular error as well as convergence speed on these established benchmarks as well.
>
> *We greatly appreciate your feedback, and please let us know if these changes have addressed your concerns and if there are other questions you may have!*
>
>
> **Zip File:**
>
> /attachment/ca0da80e186d36f18406a1fb3b77c9d6ab2e9209.zip